# Dissipative Soliton Mode-Locked Erbium-Doped Fiber Laser Using Nb_2_AlC Nanomaterial Saturable Absorber

**DOI:** 10.3390/nano13050810

**Published:** 2023-02-22

**Authors:** Arni Munira Markom, Nurul Athirah Mohamad Abdul Ghafar, Malathy Batumalay, Yusrina Yusof, Ahmad Haziq Aiman Rosol, Nur Farhanah Zulkipli, Ahmad Razif Muhammad, Hazlihan Haris, Ismail Saad, Sulaiman Wadi Harun

**Affiliations:** 1School of Electrical Engineering, College of Engineering, Universiti Teknologi MARA, Shah Alam 40450, Selangor, Malaysia; 2Faculty of Data Science and IT, INTI International University, Nilai 71800, Selangor, Malaysia; 3Department of Electronic Systems Engineering, Malaysia–Japan International Institute of Technology, Universiti Teknologi Malaysia, Jalan Sultan Yahya, Kuala Lumpur 54100, Selangor, Malaysia; 4Department of Engineering and Built Environment, Tunku Abdul Rahman University College (TARUC) Penang Branch Campus, Tanjong Bungah 11200, Pulau Pinang, Malaysia; 5Institute of Microengineering and Nanoelectronics (IMEN), Universiti Kebangsaan Malaysia (UKM), Bangi 43600, Selangor, Malaysia; 6Faculty of Engineering, Universiti Malaysia Sabah (UMS), Kota Kinabalu 88400, Sabah, Malaysia; 7Department of Electrical Engineering, University of Malaya, Kuala Lumpur 50603, Selangor, Malaysia

**Keywords:** mode-locked, fiber laser, soliton, MAX phase saturable absorber

## Abstract

We report the fabrication of an erbium-doped fiber-based saturable absorber (SA) of niobium aluminium carbide (Nb_2_AlC) nanomaterial that can generate a dissipative soliton mode-locked pulse. Stable mode-locked pulses operating at 1530 nm with repetition rates of 1 MHz and pulse widths of 6.375 ps were produced using polyvinyl alcohol (PVA) and the Nb2AlC nanomaterial. A peak pulse energy of 7.43 nJ was measured at 175.87 mW pump power. In addition to providing some useful design suggestions for manufacturing SAs based on MAX phase materials, this work shows the MAX phase materials’ immense potential for making ultra-short laser pulses.

## 1. Introduction

The presence of stimulated emission has been known since 1917, long before anyone had considered this possibility. Townes, Gordon, and Zeiger were the first to apply this concept in 1953, when they built a microwave amplification by stimulated emission of radiation (maser) [1]. Then, a few years later, light amplification by stimulated emission of radiation (laser) was invented. Another significant development was the creation of the first laser in 1960 at Hughes Laboratories using cube-shaped ruby crystals [2]. Then, a spectacular era of laser development had just begun.

The term “fiber lasers” refers to modern laser technology that is created and developed from a single optical fiber. Low loss, uniformity uniform, and high optical gain in a large communication transmission range, which enable the use of long optical fiber across thousands of kilometres in telecommunications, were the driving forces behind the development of fiber lasers. Today, ultrafast fiber lasers are well established in a variety of applications, including optical communications [3], optical sensors [4], microscopy [5], micromachining [6], laser surgery [7], and dermatology [8], which have grown into lucrative markets for international corporations.

Saturable absorbers (SA), a particular substance that can be one-dimensional (1D), two-dimensional (2D), or three-dimensional (3D), help to generate ultrashort optical pulses with a high repetition rate of frequency. Carbon nanotubes (CNTs) are 1D materials that are inexpensive, simple to fabricate, and have the definite advantage of having a high damage threshold [9]. However, non-saturable losses restrict the growth of CNTs. Graphene has received the greatest attention in the study of 2D materials in order to overcome the limitation of CNTs. A material with extremely high electron mobility and broad-band optical absorption characteristics, graphene is a band-free semi-metallic semiconductor [10]. Still, the non-gap energy band structure restricts graphene’s use and advancement in the optoelectronics industry.

SAs can also be made of 3D materials, including nanoscale charcoal powder, metal nanospheres, and Dirac semimetal cadmium arsenide (Cd_3_As_2_) [11]. Nevertheless, the significant insertion loss limits additional uses in ultrafast pulsed lasers because the majority of 3D materials do not have the saturable absorption feature. Although both 2D and 3D materials may have excellent SA properties, 3D SAs are less likely to be worth investigating due to their lack of controllability of atomic layer thickness and bandgap structure when compared to 2D materials [12]. Moreover, due to a number of advantages, such as the small insertion loss of the material at atomic depth, which is practical for optoelectronics, and the fact that electronic properties depend on structural elements such as doping, efficiency, density, and the ease of stacking to form multilayer structures, 2D material is becoming more common for the manufacture of SA than other materials [13].

Topological insulators [14], transition metal dichalcogenides [15], black phosphorus [16], antimonene [17], perovskites [18], and MXenes [19] are the most studied 2D materials. SAs materials must adhere to strict criteria, such as a high damage threshold, nonlinearity, small optical loss, and quick response time, in order to produce an excellent pulsed laser with a short pulse width and strong pulse energy. As a result, the quest for a suitable SA has continued over the years.

Two-dimensional MAX materials gained attention due to their unique electrical, thermal conductivities, physical, and chemical properties, which make them suitable candidates to be used in photonics technology [20,21,22]. The MAX phases are polycrystalline nanolaminates of ternary carbides and nitrides, or Mn + 1AXn (n = 1, 2, or 3) in which M is a transition metal, A is an A-group element, and X is carbon and/or nitrogen [23]. Layers of the A-group element are sandwiched between layers of Mn + 1Xn to form the hexagonal structure. Due to their unusual combination of characteristics, which necessitates thin-film synthesis for production, these materials have potential for applications including electrical connections and protective coatings.

As a result of their distinctive structure and high surface-to-volume ratio, 2D nanomaterials are one of the most intriguing fabrication methods for SA in this context [24]. They also enable the most up-to-date ultrafast fiber laser. Therefore, using the MAX phase family of Nb_2_AlC, where Nb is the transition metal, Al is the group element, and carbon serves as a SA, we demonstrate the first dissipative soliton mode-locked fiber laser in this paper. 

## 2. Nb_2_AlC Preparation and Characterization

MAX-PVA was created by combining polyvinyl alcohol (PVA) (Sigma-Aldrich, Subang Jaya, Malaysia) and Nb_2_AlC nanomaterials powder (Sigma-Aldrich, Malaysia). Due to its good film-formability, tensile strength, simplicity of emulsification, and high solubility in water, PVA has been shown to be an effective host polymer that makes thin film production possible. In addition to its approximately 85% optical transmission spectrum [25], PVA is also a desirable host material because it can survive powerful laser light in the experiment configuration and has a high melting temperature (200 °C) compared to polymethyl methacrylate (PMMA) (160 °C), while polyethylene oxide (PEO) has a much lower melting point of 67 °C. Deionized (DI) water was used to dissolve the PVA solution’s powder.

The preparation steps are shown in Figure 1. First, 100 mL of DI water was added to a beaker along with 1 g of PVA powder after it had been weighed using an electronic scale. On a hot plate, the mixture was swirled for 24 h at ambient temperature at a speed of 650 rpm (CORNING, PC400D). The Nb_2_AlC solution was then prepared by weighing Nb_2_AlC powders on an electronic balance. In a beaker, 30 mg of Nb_2_AlC powder was combined with 30 mL of DI water. The liquid was then agitated for 72 h at 350 rpm with a magnetic stirrer until it was completely diluted. Next, a thin film of Nb_2_AlC-PVA was created by combining 2.5 mL of the prepared Nb_2_AlC solution with 2.5 mL of the PVA solution. The liquid was magnetically agitated until it achieved homogeneity. To form the Nb_2_AlC composite film, the Nb_2_AlC-PVA solution was carefully poured into a mould to avoid air bubbles and allowed to dry at room temperature for more than 48 h.

A Nb_2_AlC-PVA thin film with a diameter of roughly 33.8 cm was peeled from the petri dish, as illustrated in Figure 2a,b. The modification in thin film thickness could not be accomplished since it is limited by the supplied petri dish, which produces a thickness of about 30 µm. Making a practical thin film will be challenging if the solution concentration is changed since the resulting film will either be too thin or too thick, making it challenging to peel. The Nb_2_AlC-PVA thin film is mounted to FC/PC fiber-ferrules after a 1 mm cut, as shown in Figure 2c. A fiber-surface ferrule was pre-coated with an index matching gel in order to attach the SA and prevent a parasitic effect on the laser’s performance.

The optical properties of PVAs were investigated in order to confirm their MAX-potential for pulse production. First, a configuration similar to that in the inset image of Figure 2 was used to measure the thin film’s linear absorption capacity. Onto a pair of FC/PC fiber-ferrule, a broadband white light source (YOKOGAWA, ANDO AQ-4303B, Tokyo, Japan) was launched. An optical spectrum analyzer (YOKOGAWA, AQ-6370D, Tokyo, Japan, 0.02 nm resolution) with a resolution of 2.0 nm and a video bandwidth (VBW) of 10 Hz set at high sensitivity was linked to one fiber-ferrule. Figure 3a shows how a flat spectrum was recorded between the wavelengths of 900 and 1650 nm. The working wavelength of 1530 nm showed an absorption of about 6.35 dB marked with a vertical bar and red arrow in Figure 3a. A small dip around 1100 nm with lower absorption of about ~1 dB suggests that the fabricated SA might not be as excellent as at 1530 nm. 

The nonlinear absorption spectrum is depicted in Figure 3b. MAX-PVA has a saturable intensity of 0.08 MW/cm^2^, a non-saturable absorption of 64%, and a saturable absorption, also known as modulation depth, of 28%. In addition, by modifying the MAX-PVA thickness, the modulation depth and non-saturable absorption may be improved. Non-saturable absorption is highly dependent on SA thickness ratios, with thicker SA films increasing non-saturable absorption percentage and having a negative correlation with a modulation depth value. Furthermore, when a thick SA film comes into contact with a fiber ferrule, it causes a high coupling loss, which explains the phenomenon of modulation depth depletion and high non-saturable absorption [26]. 

It is worth noting that the purpose of PVA film is to provide a better interface state [27], which is critical for the performance of a SA when compared to standalone materials, for example, the topological insulator-type saturable absorbers. As a result, in response to the reviewer’s concern, the non-saturable loss can be reduced by thinning the SA film thickness while attaining the smooth and homogeneous distribution of Nb_2_AlC nanoparticles. The threshold pump power of the pulsed laser was determined using these SA characteristics [24]. Therefore, reducing the MAX-PVA thickness may lessen the loss in the laser experiment configuration, causing a lower pump power threshold and a broader operational range.

FESEM and EDX are used to analyse the surface morphology of the Nb_2_AlC nanoparticles and the elemental composition of the Nb_2_AlC. Figure 4a shows that the high yield of micron-sized layered crystals is evident in the FESEM picture of the Nb_2_AlC nanoparticles at 30,000 magnification. Niobium (Nb), aluminium (Al), carbon (C), and oxygen (O) are the four principal elements that have been identified. The general formula for MAX phases, Mn + 1AXn (MAX), with n ranging from 1 to 3, is equal to the elements Nb, Al, and C. In this instance, n = 1 and M = Nb, A = Al, and X = C. A non-metal element, M is an early transition metal, and X is either carbon or nitrogen. Sharp contrasts and clear definitions of all the elements were observed. According to additional EDX examination, the material contains the elements Nb, Al, C, and O, as shown in Figure 4b. Low levels of oxygen (O) present is due to the water film adhering to the surfaces during the process of transferring the SA film to the SEM specimen stubs; however, this has no impact on the materials’ functionality. Meanwhile, element mapping confirms the existence of Nb_2_AlC nanoparticles, as shown in Figure 4c.

## 3. Experimental Configuration

In this study, mode-locked pulses are produced in an erbium experiment configuration using a newly created SA film based on Nb_2_AlC. The laser configuration is shown in Figure 5. An optical isolator, a 980/1550 nm Wavelength Division Multiplexer (WDM), an Erbium-Doped Fiber (EDF) serving as a gain medium and absorbing at 90 dB/m at 980 nm, a SA device, and an 80/20 output coupler make up the experiment configuration. An EDF is pumped through the WDM by a 980 nm laser diode. The EDF and the Nb_2_AlC SA are separated by an isolator to ensure the oscillating laser’s unidirectional propagation in the ring laser experiment configuration.

An optical spectrum analyzer (OSA, Yokogawa AQ6370C, Tokyo, Japan) with 0.02 nm resolution is used to record the fiber laser’s spectrum from the output of an 80:20 optical coupler. Anritsu MS2683A 7.8 GHz RF spectrum analyzer and a high-speed photodetector linked to an oscilloscope (GWINSTEK: GDS-3332, 500 MHz Bandwidth, Seoul, Republic of Korea) are used to examine the output pulse train and determine the repeatability of the pulse laser. The average laser power is measured using an optical power meter (Thorlabs PM 100D, Newton, NJ, USA) linked to an InGaAs power head operating between 800 and 1700 nm (Photodiode Power Sensor S145C Integrating Sphere, Newton, NJ, USA). A 200 m-long length of SMF-28 fiber is put into the experiment configuration to alter its dispersion characteristics and increase its nonlinearity to achieve mode-locking. About 213.5 m is the total length of the laser experiment configuration.

## 4. Results and Discussion

There is no mode-locked pulse train visible in a Q-switching experiment. This is because mode-locking requires that the experiment’s configuration dispersion and nonlinearity be optimized. To achieve this, a 200 m long SMF-28 should be inserted into the experiment configuration to account for the experiment configuration dispersion. A lengthy SMF was added, which increased nonlinearity and caused spectrum expansion as well as the production of steady mode-locked pulses. Through the SA’s ability to absorb light, which causes the experiment configuration to modulate loss and change the center wavelength, the mode-locked operation was started. In order to prevent noise from building up inside the ring experiment configuration, the SA was also installed there, locking in the steady self-started state.

At an incident pump power of approximately 144.57 mW, a dissipative soliton’s single wavelength mode-locking output is established; the stable condition can be sustained up to a pump power of 175.87 mW. The single-wavelength mode-locked dissipative soliton state at 175.87 mW pump power is seen in Figure 6. The spectrum of a dissipative single-wavelength soliton is seen in Figure 6. Steep edges in the spectrum are a hallmark of the dissipative soliton in all-normal-dispersion fiber lasers [28]. The two distinct steep-edge spectra have center wavelengths of 1528.56 nm and 1532.16 nm, respectively. About 3.6 nm is the spectral edge-to-edge bandwidth.

Figure 7 depicts the temporal performance of the produced mode-locked pulse fiber laser. The temporal performance of the produced mode-lock is shown in Figure 7. It should be noted that the pulse shapes shown are constrained by our oscilloscope’s 500 MHz bandwidth. In less than 50 ms, a few harmonics were found at a pump power of 175.87 mW. The experiment’s approximate length of 213.5 m and its repetition rate of 1 MHz matched up perfectly. The pulse’s amplitude varied noticeably because of the temporal jitter that existed inside the laser experiment configuration. The fiber laser’s pulse energy is constrained by the optical fiber’s core’s small diameter, the long experiment configuration length, and photon confinement. This impact on the produced pulses may be lessened by further decreasing the experiment configuration length.

The autocorrelation trace of an ultrashort, mode-locked pulse at a maximum practicable pump power of 175.87 mW is then shown in Figure 8. A pulse with an ultrashort duration and a full-width half maximum (FWHM) of 3.68 ps was calculated and recorded by the autocorrelator. Mode-locked generation in an ultrashort time domain is clearly confirmed by the trace of sech2 fitting, which fits with the experimental data.

An RF spectrum analyzer was also used to assess the consistency and the stability of pulsed fiber laser as shown in Figure 9. The mode-locked RF spectrum has a signal-to-noise ratio (SNR) of 57.54 dB at a fundamental frequency of 1 MHz. The RF spectrum with various harmonics was recorded using a broad frequency spread of 250 MHz. This pulse laser’s temporal behavior supports the construction of a mode-locked narrow pulse width and points to a low amount of signal distortion inside the laser experiment configuration. The RF spectrum generated may be improved by improving the quality of SA and experiment configuration structure to minimize/reduce or lessen the loss.

Figure 10 displays a graph of average output power and pulse energy versus pump power. The fact that the output power and pulse energy climb practically linearly with the pump power is a distinguishing feature of a pulsed laser. Boosting the pump’s power from 144.57 to 175.87 mW increases its output power from 6.1 to 7.2 mW. The average output power against the pump power fitting line also showed a 3.3% optical-to-optical efficiency. The low optical-to-optical efficiency was due to the loss introduced with the presence of ~30 µm thickness SA thin film in the configuration. One may improve the laser’s slope efficiency by thinning the SA film; however, this causes an additional concern such as a handling issue and the SA thin film might not sustain high luminescent light circulated in the configuration. Moreover, the fabricated SA film with a ring-configuration laser system successfully generates a dissipative soliton mode-locked fiber laser. 

Pump powers in the same range as pulse energies between 6.3 and 7.43 nJ were displayed on the same graph. The high output power and pulse energy of MAX-PVA as SA are due to its low insertion loss and intra-experiment configuration loss. Figure 11 shows the peak power vs. pump power graph. The peak power value ranged from 0.87 to 1.03 kW, while the pump power varied between 144.57 and 175.87 mW.

Ultimately, our dissipative soliton mode-locked fiber laser’s output performance was compared to that of comparable MAX-phase mode-locked fiber lasers. The results are summarised in Table 1. To date, no reports on Nb_2_AlC-SA generating a dissipative soliton mode-locked laser exist. The output mode-locked pulses from our laser exhibited the broadest temporal band despite the experiment configuration dispersion’s imperfect tuning. More experiment configuration adjusting is primarily necessary.

## 5. Conclusions

An EDF laser possesses dissipative soliton mode-locking thanks to the use of a saturable absorber made of a novel MAX phase material called Nb_2_AlC. Integration of the SA into the EDF laser experiment configuration resulted in a pulsed output with a pulse width of 6.375 ps at a maximum pump power of 175.87mW. The output spectrum has a 1059.7 nm center and 3.6 nm spectral edge-to-edge bandwidth. The generated pulses also had a maximum pulse energy of 7.43 nJ, a maximum pulse power of 1.03 kW, and a maximum average output power of 7.2 mW. The slope efficiency of the EDF laser is 3.3%.

## Figures and Tables

**Figure 1 nanomaterials-13-00810-f001:**
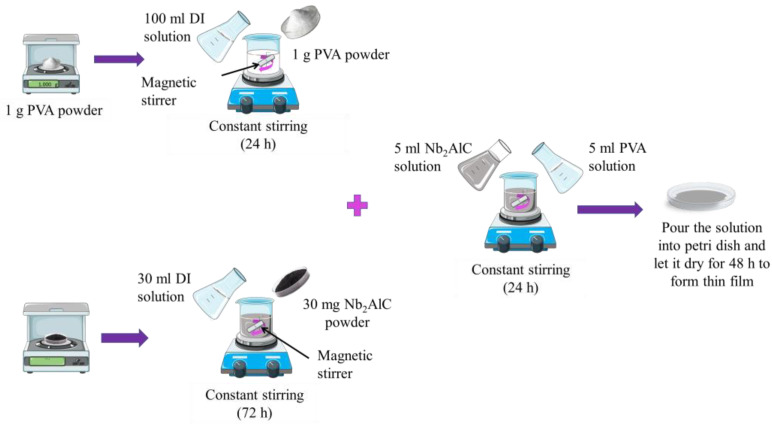
Nb_2_AlC-PVA preparation.

**Figure 2 nanomaterials-13-00810-f002:**
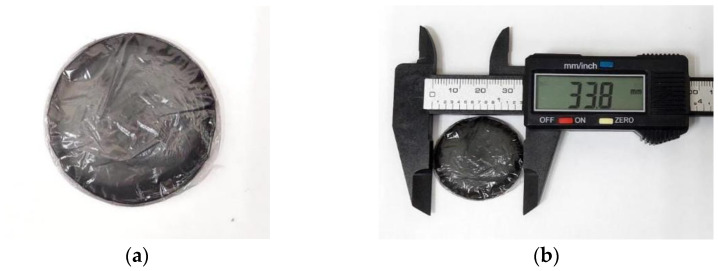
(**a**) Peeled Nb_2_AlC film (**b**) the measured Nb_2_AlC film diameter (**c**) the fabricated Nb_2_AlC film attached onto a fiber ferrule.

**Figure 3 nanomaterials-13-00810-f003:**
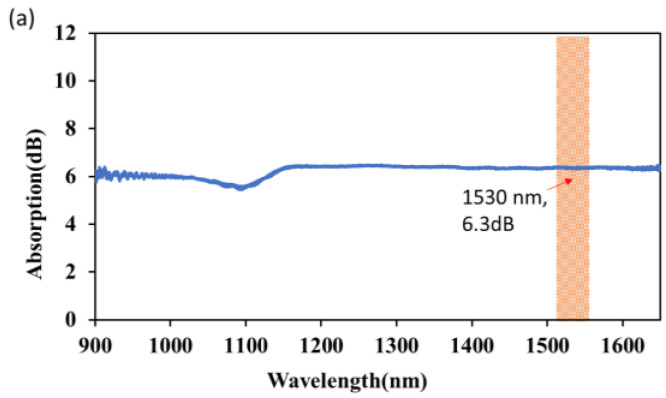
(**a**) Linear absorption of the fabricated Nb_2_AlC film with vertical bar at 1530 nm for the operating wavelength (**b**) Nonlinear absorption spectrum of MAX-PVA.

**Figure 4 nanomaterials-13-00810-f004:**
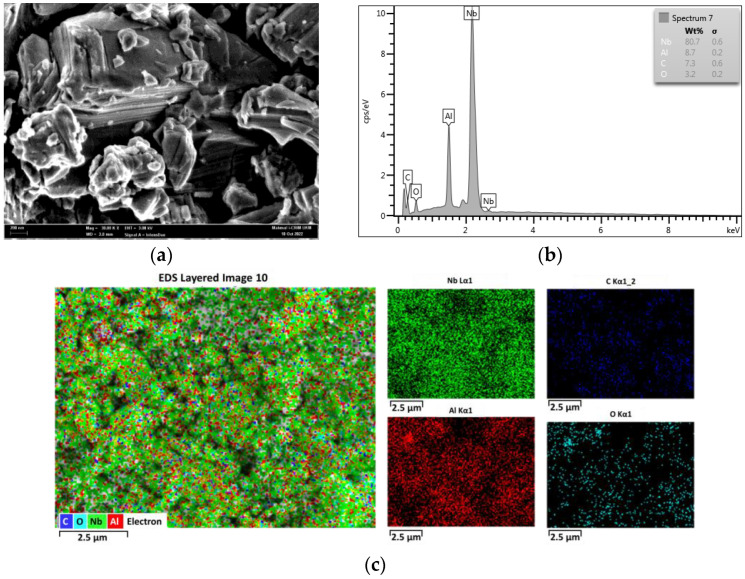
Characterisation of Nb_2_AlC (**a**) FESEM image of Nb_2_AlC-PVA film at 200 nm scale (**b**) EDX analysis with Nb, Al, C, and O elements (**c**) Mapping of Nb_2_AlC nanoparticles.

**Figure 5 nanomaterials-13-00810-f005:**
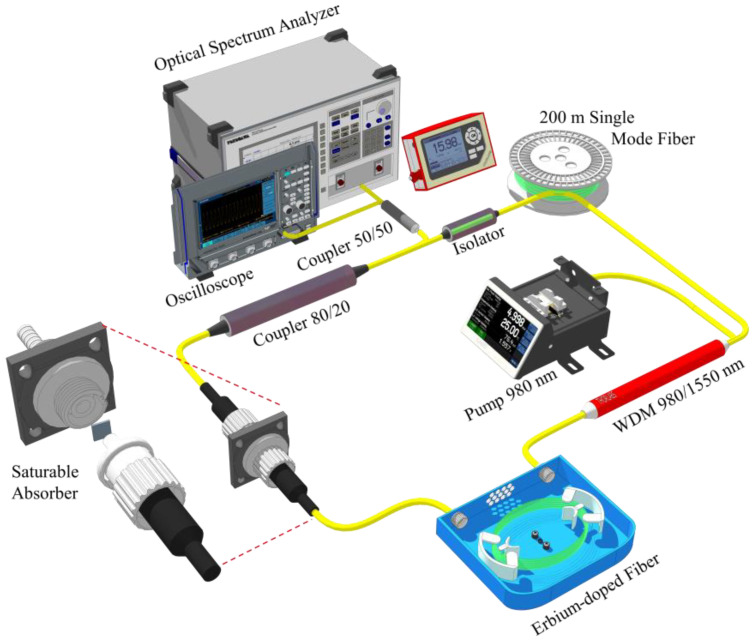
Schematic of the mode-locked fiber laser utilizing Nb_2_AlC-PVA SA with the addition of 200 m long SMF.

**Figure 6 nanomaterials-13-00810-f006:**
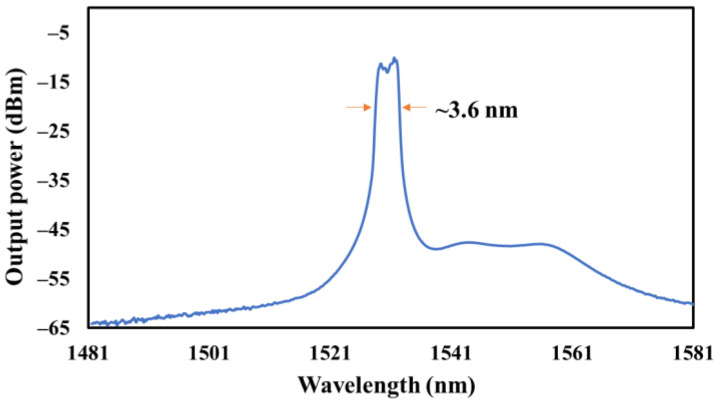
Output spectrum of dissipative soliton mode-locked MAX at 175.87 mW with 3.6 nm spectral edge-to-edge bandwidth.

**Figure 7 nanomaterials-13-00810-f007:**
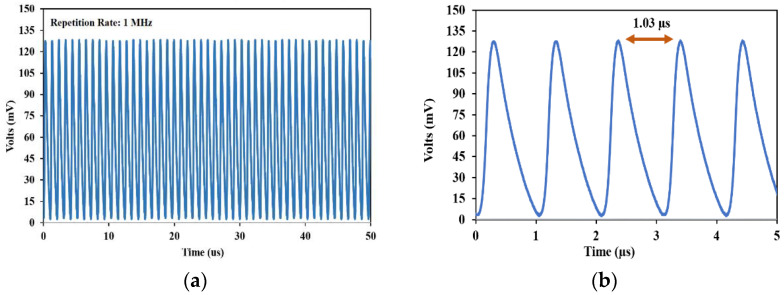
Oscilloscope pulse train at a pump power of 175.87 mW: (**a**) time-span of 50 µs (**b**) close-up pulse train at 5 µs time-span with 1.03 µs for each interval.

**Figure 8 nanomaterials-13-00810-f008:**
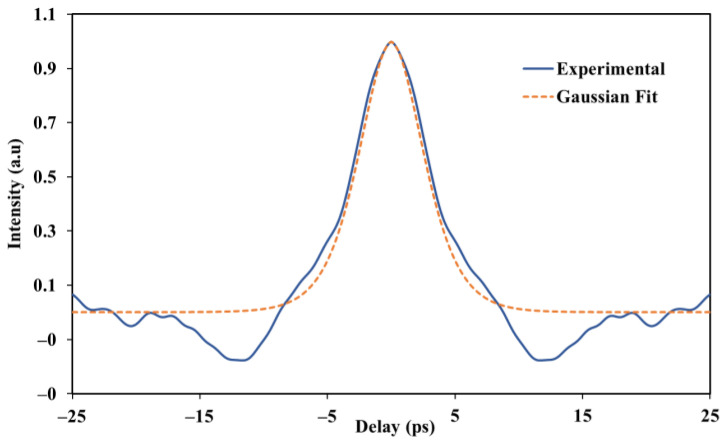
Autocorrelation output pulse trace at 175.85 mW pump power with time-span of 50 ps.

**Figure 9 nanomaterials-13-00810-f009:**
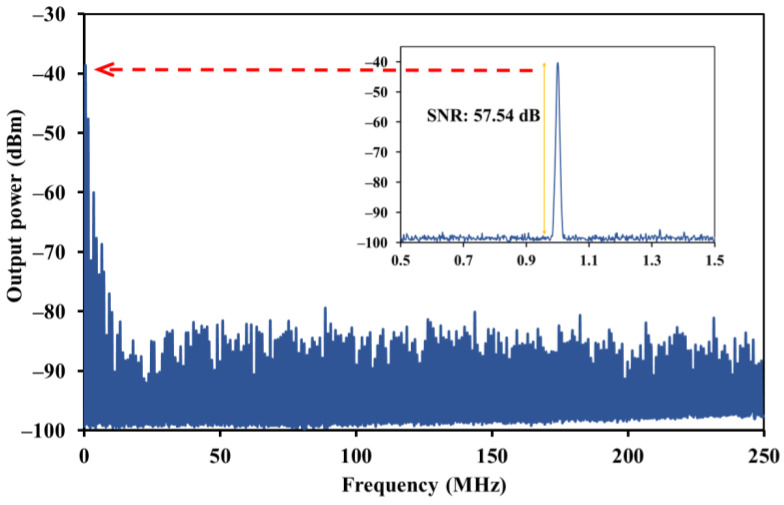
RF frequency spectrum with a 250 MHz span, signal-to-noise ratio (SNR) of 57.54 dB (inset).

**Figure 10 nanomaterials-13-00810-f010:**
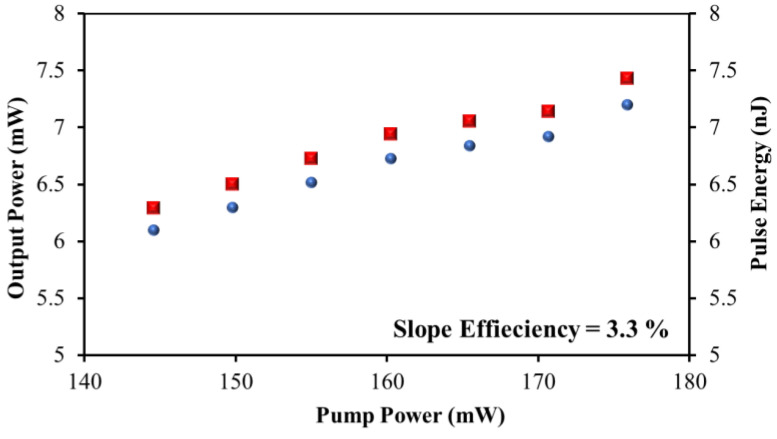
Output power and pulse energy of mode-locked MAX within 144.57 to 175.87 mW.

**Figure 11 nanomaterials-13-00810-f011:**
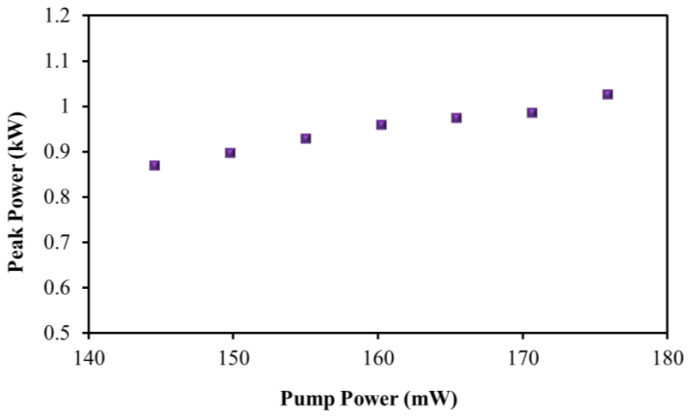
Peak power of mode-locked pulse.

**Table 1 nanomaterials-13-00810-t001:** Overview of SA based on MAX phase materials on EDF configuration.

SA	Integration Method	Center Wavelength (nm)	Threshold (mW)	Repetition Rate (MHz)	Pulse Energy (nJ)	Pulse Width (ps)	Type of Pulse	Ref.
Ta_2_AlC	Tapered	1937	245	10.73	-	1.678	Soliton Mode-locked	[28]
	Side polished fiber (SPF)	1931	351	9.52	-	1.743		
	Arc-shaped	1929	380	10.16	-	1.817		
Ti_2_AlC	Thin film	1559	220	5.16	-	0.68	Soliton Mode-locked	[29]
Ti_2_AlN	Thin film	1557.5	150	14.8	16.2	5.04	Mode-locked	[30]
Cr_2_AlC	Thin film	1559	121.69	1	0.91	4.45	Soliton mode-locked	[31]
Ti_3_AlC_2_	Thin film	1557.77	103.6	1.887	8.16	3.68	Mode-locked	[32]
Ti_3_AlC_2_	Thin film	1559.7	66.3	1.8	4.46	5.02	Soliton mode-locked	[33]
Ti_3_AlC_2_	D-shaped	1557.63	21	1.89	8.14	2.21	Soliton mode-locked	[34]
Nb_2_AlC	Thin-film	1987.2	10340/10.3 W	0.135	11540/11.54 µJ	850000/850 ns	Q-switched	[35]
Nb_2_AlC	Thin film	1530	144.57	1	7.43	6.375	Dissipative soliton mode-locked	This work

## Data Availability

Not applicable.

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
