# Peer review of "Dissipative Soliton Mode-Locked Erbium-Doped Fiber Laser Using Nb2AlC Nanomaterial Saturable Absorber"

_nanomaterials, 2023, doi:10.3390/nano13050810_

Round 1

Reviewer 1 Report

This paper demonstrates a mode locked fiber laser using Nb2AlC as saturable absorber. The manuscript is well drafted with detailed experimental procedure and convincing result. I would recommend publication after a minor modification.

In page 3 last paragraph, the authors mentioned the challenges to make a thinner or thicker film. Is this limited by the surface of the petri dish? What’s the thickness of the film produced in this experiment? 

Reviewer 2 Report

The paper is of interest but I have many question about certain statements in the manuscript.  I made attempts to restate some of the unclear sentences or phrases.  I believe the paper could be worth publishing once clarifications are made to the paper.

Round 2

Reviewer 2 Report

Item 4, A century is 100 years.  1953 to 1960 is only 7 years.

Many of my comments and questions are answered in the response letter but not in the text of the paper.  Example is Item 5 where I didn't understand what 'transmission area' meant.  It should have been changed to communication transmission range. 

Item 14, 100% transmission must refer to a specific spectral range.  There are likely ultraviolet and infrared limits which should be specified.

Item 19, more of the response to my question should appear in the text.  Also, I am not convinced that increasing the frequency of the modelocking source will change the non-saturable loss.

Item 25, The 500 MHz bandwidth should tell you that the pulse shapes shown in Fig 8 are limited by your detection system and you should tell the reader that too.  Also the term 'resolution' in Fig 8 is used incorrectly. Maybe use time-span of 50 us , 5 us. Same resolution issue for Fig 9.

Item 32, response regarding how to improve the poor 3.3% efficiency should be included in the paper. 

English remains awkward and confusing in many places.  I see no evidence that your colleague made any English changes.
